# Encouraging Adults Aged 65 and over to Participate in Resistance Training by Linking Them with a Peer: A Pilot Study

**DOI:** 10.3390/ijerph20043248

**Published:** 2023-02-13

**Authors:** Elissa Burton, Keith D. Hill, Jim Codde, Angela Jacques, Yoke Leng Ng, Anne-Marie Hill

**Affiliations:** 1Curtin School of Allied Health, Curtin University, Perth, WA 6102, Australia; 2enAble Institute, Curtin University, Perth, WA 6845, Australia; 3Rehabilitation, Ageing and Independent Living (RAIL) Research Centre, Monash University, Frankston, VIC 3199, Australia; 4National Centre for Healthy Ageing, Monash University and Peninsula Health, Frankston, VIC 3199, Australia; 5Institute for Health Research, University of Notre Dame Australia, Perth, WA 6160, Australia; 6Health and Social Sciences Cluster, Singapore Institute of Technology, Singapore 138683, Singapore; 7School of Allied Health, The University of Western Australia, Perth, WA 6000, Australia

**Keywords:** strength training, peer support, exercise, adherence

## Abstract

Resistance training (RT) is beneficial for older adults, particularly to support living independently in their home. However, fewer than 25% of older adults in Australia participate in the recommended, twice-weekly sessions. Reasons older adults do not participate in RT include not having anyone to go with or not knowing what to do. Our study linked older adults with a peer (i.e., older person already participating in RT) to help them overcome these barriers. The aim of our study was to determine whether peer support was suitable for older adults participating in RT for the first time in the home or gymnasium setting. Each group (home vs. gymnasium) received a 6 week, twice-weekly program. Twenty-one participants completed the 6 week intervention: 14 in the home group and seven in the gymnasium group. The home group completed significantly more sessions per week (2.7 vs. 1.8) than the gymnasium group. Although both groups significantly improved on many physical assessments, no between-group differences were found. However, it is suitable to link a peer for support with novice older people participating in a RT program for the first time in the home or gymnasium. It is recommended that future studies explore whether peer support improves sustainability.

## 1. Introduction

Most older adults want to live in their own home for as long as possible. To achieve this, they need to maintain their health, be able to complete essential activities of daily living (ADLs) such as showering, toileting, dressing, transfers, and cooking, be able to participate in enjoyable activities, such as social outings, to preserve autonomy, competence, and quality of life [1]. Staying physically active is important for living independently and maintaining good health and quality of life. The World Health Organization (WHO) recommends older adults undertake regular physical activity, which should include participating in at least 150 min of moderate aerobic activity per week (≥75 min of vigorous activity), two or more sessions of resistance training (RT), and three or more sessions of balance training per week to assist in the prevention of falls [2].

There are numerous benefits to participating in RT on a regular basis. RT can assist in improving physical function, muscle strength, power, bone density, and mental health, as well as relieve pain [3,4,5]. It can also improve mobility and performance in ADLs, reduce the risk of frailty or sarcopenia, and preserve independence [6,7]. Although the evidence for the benefits of RT for older adults is high, Bennie and colleagues reported that meeting RT recommendations (i.e., at least twice a week) is undertaken by fewer than one-quarter of older adults in America and even fewer in Australia [8].

There are many barriers for older people participating in RT; these can include physical, psychological, social, and environmental factors [9,10]. Lacking social support, having no one to go with, not knowing where to go to participate, or not having a suitable program near home are all barriers facing older adults when they consider participating in RT [9,10]. Peer support may be one way to remove the barrier for older adults who have no one to go with, or they potentially do not have someone who is supportive of them undertaking a new activity, such as RT [9,10]. Families may be concerned with their older relative undertaking this type of activity and, therefore, may not be supportive, trying to persuade them to take up less active options. Peers are characterized by having similar characteristics, such as age, education, family status, or even religious beliefs [11]. Peers usually have experience in the area of interest and undertake training in this specific area to avoid going outside the parameters of a project [11]. A strength of including peers is that they have a capacity to relate, share their past experience, and empathize with their target group or person they are linked to, which nonpeers may not be able to [12].

A number of studies have investigated the effects of using peer support [13,14,15,16], peer mentors [17,18,19,20], or peer leaders to deliver physical activity interventions [21,22,23,24,25,26]. Many included a mix of endurance, flexibility, strength, and balance training within the intervention [13,14,17,18,19,21,22,23,26,27,28], with an additional study exploring aquatic exercise [29]; two studies used walking only interventions [15,16], and one utilized only strength and balance training and was peer-led [25]. The findings across these various studies suggest that there are benefits to having a peer involved, such as reporting increased levels of physical activity and improvements in physical function, although, when combining studies into a meta-analysis, the control group improved more for both walking and handgrip compared to the intervention groups [11]. A more recent 12 week peer-led exercise intervention by Bouchard and colleagues reported improvements for the intervention group compared to controls for the chair stand test, arm curl, and the timed up and go (TUG) [21]. However, it should be noted that this study included participants 50 years and older as opposed to the others described above that had an inclusion criterion of 60 years and over [11]. It is interesting to note how few studies have focused on strength and balance training only, when it is well known that this type of training is the most effective in reducing falls [30], which can be devastating to the older person involved and costs health systems across the world millions of dollars every year [31]. Furthermore, it is well established that older adults’ preferred type of exercise or physical activity is walking, and that, as mentioned earlier, they are more likely not to meet the physical activity guidelines for balance and RT training [8].

To avoid the barriers described above and provide a positive experience for older adults who have not participated in RT prior to this study, a peer was linked with each RT participant. It is also unclear whether older people would participate more frequently and adhere to a program more if delivered in the home or at a gymnasium with state-of-the-art equipment, given that little data are available on the adherence of physical activity programs that involve peers [11].

Therefore, the aim of this study was to determine whether peer support was suitable for older adults participating in RT for the first time in the home or gymnasium setting and, in doing so, determine recruitment and dropout rates and whether adverse events occurred, while participating in the exercise interventions, to assist with future definitive studies. A secondary aim was to determine whether there were differences between those training at home compared to those training in the gymnasium setting.

Experiences of being paired with a peer and interactions between the exercise participants and their peers have been published elsewhere [32,33]. These are not described in this article.

## 2. Materials and Methods

### 2.1. Design

This was a pilot study with two intervention groups: (1) RT at a gymnasium; (2) RT at home. Participants in both groups were all connected with a peer (see Section 2.4 below for further details). Each RT intervention ran for 6 weeks.

### 2.2. Participants and Setting

Recruitment occurred through flyers being placed in shopping centers and local retirement villages, as well as snowballing with friends or neighbors, who were asked if they would also like to participate. Peers were recruited through the local RT programs, word of mouth, and flyers at the local retirement villages. Inclusion criteria for both RT programs were being aged 65 years and over, not currently participating in RT, being able to speak and understand English, not having a diagnosis of dementia, and having no medical conditions preventing them from participating in the RT program. People recruited into the gymnasium group were already physically active (i.e., >150 min of moderate physical activity per week by self-report), but not participating in RT, and they either participated in the group sessions together with their peer or contacted their peer by phone. People recruited into the home-based RT group met the same general inclusion criteria above, but were not physically active (i.e., ≤150 min of moderate physical activity per week, by self-report). Participants in the home-based RT group had contact with their peer by phone and/or face to face in a social setting (e.g., the local café). Peers were included if they were currently participating in a RT program (for at least 2 months).

### 2.3. Ethical Considerations

The study received ethics approval from two Universities, Curtin University (HRE2017-0259) and the University of Notre Dame (Australia) (016155F). All participants provided written informed consent prior to any data collection commencing.

### 2.4. Peer Training

Peers completed a 3 h training session prior to the RT participants commencing in their intervention programs. A full account of the peer training has been published previously [32,33]. The peer training included defining the peer and the peer role, presenting reasons why older adults participate or do not participate in RT (i.e., motivators and barriers), communicating with others, working as a team, and how to be a role model. The relationship between the RT participant and their peer was intentionally organic, as it is understood that different people behave and react in different ways. The peers were provided with potential discussion points and ideas on how to communicate, but no script or specific tasks were expected from these communication points. The peers and their RT participants identified their preferred way to keep in contact, and it differed between completing the gymnasium sessions together, via phone call or text, or meeting up in a social setting, such as the local café for a coffee.

### 2.5. Intervention

The peers were asked to contact the RT participant they were linked to at least once per week. The type of contact (e.g., face-to-face, phone, or during RT sessions) was left up to the dyad. The home RT program was delivered by a qualified physiotherapist with long-term experience working with older adults. The physiotherapist was provided with a suite of upper- and lower-body strength exercises (upper body: *n* = 12; lower body: *n* = 12) and balance exercises (*n* = 6) to choose from, to allow for individualization per participant. Hand and ankle weights varying from 0.5 kg through to 6 kg were available, and therabands (light, medium, and heavy) were also used. The physiotherapist visited each participant once a week, for 1 h for 6 weeks, progressing the load and number of sets and repetitions as required, and encouraging the participant to complete a minimum of two sessions per week.

The gymnasium RT program was held in a wellness center. The gymnasium program was delivered by an exercise physiologist, and the program was part of a seniors (over 50 s) RT program, called Living Longer, Living Stronger (now known as Strength for Life). The participants were given a RT program at the commencement of their 6 weeks and were encouraged to attend for 1 h, twice a week for 6 weeks, to this group workout. The gymnasium participants had a choice of six different sessions they could attend each week. The gym participants completed an average of four lower body strength exercises, five upper body strength exercises, one core exercise, and two balance exercises. There is usually a cost to participate in this program; to avoid that being a barrier to participation, the research study paid for session attendance. The equipment used by the gymnasium participants included machine and free weights, therabands, medicine balls, and balance equipment (e.g., wobble board). The gymnasium group was also required to take a towel with them to clean the equipment after they had used it, and to wear enclosed and safe footwear. Similar to the home RT participants, their program was progressed over the 6 week intervention period.

### 2.6. Data Collection and Procedure

All of the RT participants completed data collection prior to commencing their RT program and meeting their peer. The physiotherapist completed the pre and post data collection for the home participants, while the exercise physiologist collected it for the gymnasium participants. This also allowed each health professional to determine the strength and balance levels of each participant, which made developing their RT program easier. This was a pilot study; therefore, blinding of participants nor assessors did not occur.

Measurements included demographic and health information, height and weight (calculated as body mass index (BMI)), fall history (past 12 months, self-report), functional reach [34], the 6 min walk test (6MWT) [35], sit to stand five times [36], timed up and go (TUG, comfortable speed) [37], timed tandem walk [38], and the handgrip strength test, using a JAMAR dynamometer, according to the method described by the American Society of Hand Therapists [39,40].

### 2.7. Sample Size

The sample size for this pilot study was set at 8–10 peers and 20–30 RT participants (i.e., 10–15 per RT group). Given that this was a pilot study, and that suitability not effectiveness of the intervention was the aim, a sample size calculation based on a primary (physical assessment) outcome was not appropriate to be undertaken [41]. It has been recommended that pilot studies can have between 12 to 30 subjects; hence, the sample size of 20–30 was set a priori [42,43]. It was expected that, with this number, piloting the intervention, peer training, and outcome tools could be used for calculating the sample size for a future large randomized controlled trial.

### 2.8. Analysis

Participant demographic and baseline data were summarized using frequency distributions for categorical data and means/standard deviations for continuous data. Chi-squared tests and *t*-tests were used for categorical and continuous group comparisons, respectively. Pre and post outcomes were compared within and between groups using generalized linear mixed models with random subject effects and group–time interactions. Results were summarized using marginal mean estimates and 95% confidence intervals. Stata version 17.0 (StataCorp, College Station, TX, USA) was used for data analysis, and significance levels were set at alpha = 0.05.

## 3. Results

### 3.1. Participant Characteristics

Twenty-two older adults provided consent; however, only 21 completed the intervention, with one participant withdrawing due to illness prior to commencing the intervention. Fourteen participated in the home and seven participated at the gymnasium. It was more difficult to recruit older people willing to participate in the RT program at the gymnasium than at home. The average age was 76.1 (±5.9) years, with no difference between the groups; there were eight males and 13 females who participated. Table 1 presents the demographics for all participants and within their group (i.e., home or gymnasium), excluding the withdrawal. Significantly more participants in the home intervention self-reported memory loss than the gymnasium group.

### 3.2. Suitability of Intervention

Recruitment occurred across the southern suburbs of Perth, Western Australia, including at a large retirement village. Flyers were placed at shopping centers, and presentations were undertaken with older adults at the retirement village. Participants who had agreed to take part were also asked if they had friends that might be interested. It is difficult to determine an exact recruitment rate due to these various recruitment strategies; however, 33 people expressed an interest, potentially giving a recruitment rate of 66.7%. One participant withdrew prior to commencing the intervention; the withdrawal rate was low at 4.5%. All other participants completed the 6 week interventions. No adverse events, including falls, were reported either during the intervention (to the peer or exercise instructor) or at post-intervention data collection.

### 3.3. Differences between Home and Gymnasium Training

The participants in the home training group completed an average of 16.0 (±2.2) sessions over the 6 week intervention, which was an average of 2.7 (±0.4) sessions per week. The gymnasium group attended 11.0 (±3.3) sessions on average across the 6 week intervention, which was an average of 1.8 (±0.6) sessions per week. The home training group slightly exceeded our goal of two sessions per week, with the gymnasium training group being below the recommended goal. There was a significant difference between the groups for total number of sessions completed (t(18) = 3.969, *p* < 0.001, 95% CI: 2.353, 7.647) and the weekly adherence (t(18) = 3.967, *p* < 0.001, 95% CI: 0.391, 1.274). It appears suitable and safe to conduct RT programs in the home or at the gymnasium when linking novice participants with a peer who has experience participating in RT programs. However, those training at home completed more sessions.

Table 2 presents the pre–post results within and between group. There were no significant differences between the groups for the 6 week intervention, demonstrating that it is as effective for an older adult to train at a gymnasium or at home. Significant differences from baseline to post-testing were found within both groups. BMI significantly increased within the gym training group, with no difference in the home training group. Functional reach significantly improved for the home training group but not the gymnasium group. For sit to stand, the 6MWT, and the TUG, both groups significantly improved from baseline to post-testing. It must be noted, however, that the home training group trended toward greater improvements in the objective outcome measures, even though they were limited to dumbbells and therabands within their own environment and not the additional pieces of fitness equipment available in the gymnasium.

## 4. Discussion

RT programs can be conducted safely in the home or at a gymnasium when linking novice older participants with a peer who has had experience participating in RT programs providing support. Older adults who participated in the home RT program with a physiotherapist and received peer support participated in significantly more sessions across the 6 week intervention (2.7 vs. 1.8 per week) compared to those in the gymnasium intervention with peer support. This may be because they were able to participate at a time of the day and days of the week that suited them, and they could be flexible, if needed. In contrast, the gymnasium training sessions only provided six opportunities across the week, and, if someone was sick or had other appointments, they could not attend alternate sessions easily. Given that all participants, regardless of intervention group, participated in the RT program for 6 weeks (only one withdrawal prior to commencing training), this does provide some evidence that peers may be beneficial in supporting participation in RT interventions, regardless of whether they are conducted at home or in a gymnasium group setting. This is a similar retention rate to other peer intervention studies that also successfully promoted health practices for older adults [44,45]. However, a number of the multicomponent physical activity interventions led, mentored, or supported by peers did not include adherence to the intervention; they only included retention rates [14,17,25]. It is recommended that future studies include adherence rates, as well as withdrawal/retention rates, to better understand the value of utilizing peers in physical activity interventions. Two studies by Dorgo and colleagues [19,27] reported a 75% adherence rate, and Bouchard et al. found that 68% of the intervention sessions were attended by their participants [21]. RT programs without peer support have reported withdrawal rates of up to 44% for older adults [46,47,48], indicating growing evidence of the benefits of including peers in physical activity interventions.

Although the home intervention participants were only given dumbbells and therabands of varying levels, and although they were not exposed to the many different types of equipment as gymnasium participants, they improved their strength and balance as much as the gymnasium group. The home training group also self-reported that they were not meeting the WHO physical activity guidelines prior to commencing the study, which may have been another reason for the greater improvement levels (although not significant between groups) compared to the gymnasium group. The reason those not self-reporting to be physically active were given the home intervention was due to past research showing barriers to physical activity or RT participation including aspects such as not knowing what to wear, not knowing anyone who is doing RT, and not knowing where to go and participate in a program [9]. Many of these issues were not a concern to those already self-reporting to be physically active because they already knew about the RT program at the gymnasium and knew what to wear. Their main barrier was the cost, which was removed by paying the gymnasium session fees for the 6 week intervention.

It must be noted that the cost to subsidize the gymnasium group (10 AUD per session) was markedly cheaper than the cost of having a physiotherapist weekly (50 AUD/h per participant) within the home of each home-based participant. Given that there was no difference in the strength or balance outcomes between the two groups, this needs to be taken into account when considering future studies or how they can be translated into practice, because cost is important, not only to the consumer but also to health agencies. However, the one-to-one attention that the home training group received from the physiotherapist may have been a reason as to why their adherence was significantly higher than the gymnasium group, and this needs to be considered, although it may be potentially difficult to sustain this level of support over the longer term. It must also be noted that both groups saw an increase in their BMI over the 6 week intervention; however, the gymnasium group reported a significant increase. It is unknown what the reason for this was because muscle mass and body fat levels were not assessed during this study, nor was nutrition intake over the 6 weeks.

Limitations in the study included the small sample sizes for each group. The aim of this study was to determine whether peer support could be used within these two environments (i.e., gymnasium and home) and to determine recruitment and dropout rates also for a future definitive study; this study was not powered to show effectiveness in the assessments, even though effectiveness was found in a few. It is recommended that future effectiveness studies reach an adequate sample size based on their primary outcome. Furthermore, it is worth noting that only seven participants who were already physically active were recruited to participate in the study in the gymnasium environment. Although cost can be a barrier, there are many other barriers to older people wanting to participate in RT programs within the gymnasium environment, and these need to be considered in future studies. Another limitation was having the same health professional assess and then deliver the intervention to the participants. This study was a pilot study to determine whether peer support was suitable for older adults participating in RT for the first time in the home or gymnasium setting and did not have a primary outcome investigating effectiveness. It is recommended that future studies reduce bias by having blinded assessors who differ to those delivering the intervention. Moreover, future studies should consider evaluating the differences between interventions delivered by a health professional (e.g., physiotherapist or exercise physiologist) only and those delivered by a health professional, while also including peer support for the older participants, to have a better understanding of the effect that peers may have on new participants in RT.

Studies that included peers often had them completing one of two roles: (1) where an intervention is led by a peer, or (2) where the peer provides support and motivation [11], which occurred in this study. Although the evidence remains mixed as to the effectiveness of peer-led programs [49], future studies could consider a more hands-on role for the peer, potentially after a health professional (i.e., exercise physiologist or physiotherapist) has assessed and identified the program commencement level for the older participant. Future studies could also consider how peers can be utilized to assist in sustaining participation in RT over the longer term, so that more older people are meeting the recommended physical activity guidelines for gaining long-term health benefits.

## 5. Conclusions

This study found that older adults participating in RT for the first time with peer support could meet the recommended physical activity guidelines of two sessions per week. The home training group completed significantly more total sessions and sessions per week than the gymnasium group, but there were no significant differences between the groups for the strength, balance, or endurance performance measures over the 6 week intervention. Peer support could potentially assist in reducing the number of older adults withdrawing from RT programs, and it is recommended that future studies investigate this further.

## Figures and Tables

**Table 1 ijerph-20-03248-t001:** Demographics.

Factor	Category	All	Home Exercisers	Gym	*p*-Value
Number of participants		21	14	7	NA
Age, years, mean (SD)		76.1 (5.9)	76.6 (6.6)	75.1 (4.2)	0.593
Gender, *n* (%)	Male	8 (38.1)	4 (28.6)	4 (57.1)	0.204
	Female	13 (61.9)	10 (71.4)	3 (42.9)	
Marital status, *n* (%)	Married/de facto	16 (76.2)	11 (78.6)	5 (71.4)	0.063
	Widowed	3 (14.3)	3 (21.14)	0 (0.0)	
	Separated/divorced	2 (9.5)	0 (0.0)	2 (28.6)	
Live alone, *n* (%)	Alone	5 (23.8)	3 (21.4)	2 (28.6)	0.717
	With spouse/partner	16 (76.2)	11 (78.6)	5 (71.4)	
Smoker, *n* (%)		5 (23.8)	3 (21.4)	2 (28.6)	0.717
Self-reported comorbidities, *n* (%)	CV disease	9 (42.9)	5 (35.7)	4 (57.1)	0.350
	Respiratory disease	4 (19.0)	2 (14.3)	2 (28.6)	0.432
	Spinal problems	9 (42.9)	6 (42.9)	3 (42.9)	1.000
	Osteoporosis	3 (14.3)	2 (14.3)	1 (14.3)	1.000
	Hearing impairment	6 (28.6)	3 (21.4)	3 (42.9)	0.306
	Visual impairment	17 (81.0)	10 (71.4)	7 (100.0)	0.116
	Diabetes	4 (19.0)	2 (14.3)	2 (28.6)	0.432
	Memory loss	5 (23.8)	1 (7.1)	4 (57.1)	0.011
	Recent surgery	8 (38.1)	4 (28.6)	4 (57.1)	0.204
	Falls in last 12 months	8 (38.1)	7 (50.0)	1 (14.3)	0.112
Trouble walking, *n* (%)	No	15 (71.4)	8 (57.1)	7 (100.0)	0.241
	Some but do not use aid	2 (9.5)	2 (14.3)	0 (0.0)	
	Need a stick or frame to walk outside	3 (14.3)	3 (21.4)	0 (0.0)	
	Need a stick or frame to walk inside home	1 (4.8)	1 (7.1)	0 (0.0)	

NA: not applicable, SD: standard deviation; *n*: number; CV: cardiovascular.

**Table 2 ijerph-20-03248-t002:** Pre and post outcomes by group.

Outcome	Time	Home Training			Gymnasium			
		Est Mean	95% CI	P_1_	Est Mean	95% CI	P_2_	P *
Body mass index	Pre	28.0	25.83–30.28		27.3	24.09–30.61		
(kg/m^2^)	Post	28.2	25.95–30.40	0.171	27.6	24.33–30.85	0.050	0.417
Functional reach	Pre	29.1	26.42–31.71		30.2	26.31–34.04		
(cm)	Post	31.9	29.21–34.58	0.005	32.4	28.57–36.31	0.111	0.746
Sit to stand (5 times)	Pre	12.6	11.16–14.01		16.2	14.14–18.31		
(s)	Post	10.0	8.57–11.45	<0.001	13.7	11.58–15.75	<0.001	0.995
Timed up and go	Pre	7.3	6.57–8.29		7.9	6.71–9.74		
(s)	Post	6.9	6.19–7.72	0.066	7.2	6.15–8.60	0.054	0.605
6 min walk test	Pre	442.7	386.60–498.74		441.8	359.70–523.86		
(m)	Post	474.4	417.46–531.41	<0.001	471.8	389.73–553.88	0.005	0.901
Timed tandem walk	Pre	15.8	13.07–20.02		12.9	10.43–16.97		
(s)	Post	11.1	9.64–13.04	<0.001	10.6	8.89–13.24	0.009	0.199
Grip strength	Pre	21.6	16.56–27.40		34.3	25.03–45.01		
(kg)	Post	20.6	15.56–26.27	0.566	31.2	22.38–41.42	0.345	0.659

Est: estimated; CI: confidence interval; P_1_/P_2_: within group pre–post difference; P *: group × time interaction.

## Data Availability

Data can be made available from the authors upon reasonable request.

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
