# Peer review of "Encouraging Adults Aged 65 and over to Participate in Resistance Training by Linking Them with a Peer: A Pilot Study"

_ijerph, 2023, doi:10.3390/ijerph20043248_

Round 1
Reviewer 1 Report
The topic is interesting and provide some useful information regarding the motivation for older adults in participating in resistance training for the first time in the home or gymnasium setting and the influence of peer support.
The manuscript is clear and well structured, nevertheless there are some issues that need to be improved. Therefore, a revision is required
Introduction
Page 1 line 36. The affirmation ‘Most older adults want to live in their own home for as long as possible. To achieve this, they need to be able to complete essential activities ……. ’ should be modified, including the concept of autonomy and quality of life. Presented in these terms, the message is unclear.
See:
Haynes A, Sherrington C, Wallbank G, Wickham J, Tong A, Kirkham C, Manning S, Ramsay E, Tiedemann A.Using self-determination theory to understand and improve recruitment for the Coaching for Healthy Ageing (CHAnGE) trial. PLoS One. 2021 Nov 19;16(11):e0259873. doi: 10.1371/journal.pone.0259873. eCollection 2021.
Materials and methods
Page 4 line 157. The Authors reported that height and weight were measured, but they did not mention BMI calculation, which is instead considered in the results. Please add it.
Page 4 line 164. It is not clear if the Authors have calculated the minimum recommended size for their research. They should present the power levels, using for example, the G*Power software supports sample size and power calculation for various statistical methods. It is also necessary to present the sampling error (your margin of error) and achieved the confidence level in the study. Always, in relation to the sample size it is necessary to present the sampling error, the margin of error, as well as the confidence level
Results
Page 5 line 215. The Authors stated that “Body mass index significantly increased within the gym training group”, but this aspect is not addressed in the discussion.
Discussion
In my opinion it would be interesting to also evaluate a comparison with a home resistance training program delivered by the same qualified physiotherapist, but without the support of peer, in order to better understand the real contribution of the setting or of the peer support. I know that the reasons older adults do not participate in resistance training include not having anyone to go with or not knowing what to do, but the motivation connected to being followed at home could also serve as an incentive, regardless the presence of peers. It should be added in the limitations section and in the future perspectives.
Author Response
Please note we have addressed all of the reviewer comments in the one document, please see attached.

Reviewer 2 Report
I found the presented study properly executed, well-written, and of potential interest to the readership. The study presents a novel idea that fits well in the existing knowledge about peer-mentoring in older adult fitness and therefore the study has the potential to contribute to the relevant scientific literature. The main shortcoming of the manuscript, as currently written, is its lack of proper review of the existing peer-mentor or peer-support literature for older adult fitness. The authors must update their list of references and make revisions to the Introduction and Discussion sections of the manuscript by including and synthesizing study findings from previous studies. The authors rely heavily on their own previous studies (Burton et al., 2017; Burton et al., 2018), while not considering the existing literature from other sources. As such, the current manuscript is lacking comprehensiveness and quality. I recommend the following publications from the existing literature on peer-mentoring in older adult fitness to be included:
- Bouchard, D. R., Olthuis, J. V., Bouffard-Levasseur, V., Shannon, C., McDonald, T., & Sénéchal, M. (2021). Peer-led exercise program for ageing adults to improve physical functions - A randomized trial. European Review of Aging and Physical Activity, 18(1), 2. https://doi.org/10.1186/s11556-021-00257-x
- Matz-Costa, C., Howard, E. P., Castaneda-Sceppa, C., Diaz-Valdes Iriarte, A., Lachman, M. E. (2019). Peer-Based Strategies to Support Physical Activity Interventions for Older Adults: A Typology, Conceptual Framework, and Practice Guidelines. The Gerontologist, 59(6), 1007–1016. https://doi.org/10.1093/geront/gny092
- Dorgo, S., King, G. A., Bader, J. O., & Limon, J. S. (2013). Outcomes of a Peer Mentor Implemented Fitness Program in Older Adults: A Quasi-Randomized Controlled Trial. International Journal of Nursing Studies, 50(9), 1156-1165. DOI: 10.1016/j.ijnurstu.2012.12.006
- Buman, M.P., Giacobbi, P. R., Dzierzewski, J. M., Morgan, A. A., & Marsiske, M. (2011). Peer Volunteers Improve Long-Term Maintenance of Physical Activity with older Adults. Journal of Physical Activity and Health, 8(2) S257-S266. DOI: doi.org/10.1123/jpah.8.s2.s257
- Dorgo, S., King, G. A., Bader, J. O., & Limon, J. S. (2011). Comparing the Effectiveness of Peer Mentoring and Student Mentoring in a 35-week Fitness Program for Older Adults. Archives of Gerontology and Geriatrics, 52, 344-349. DOI: 10.1016/j.archger.2010.04.007
- Dorgo, S., King, G. A., & Brickey, G. D. (2009). The Application of Peer Mentoring to Improve Fitness in Older Adults. Journal of Aging and Physical Activity, 17(3), 344-361. DOI: 10.1123/japa.17.3.344
- Dorgo, S., Robinson, K., & Bader, J. (2009). The Effectiveness of a Peer-Mentored Older Adult Fitness Program on Perceived Physical, Mental and Social Function. Journal of the American Academy of Nurse Practitioners, 21, 116-122. DOI: 10.1111/j.1745-7599.2008.00393.x
Furthermore, the manuscript could benefit from a general description in the Introduction section about the benefits of resistance training for older adults. Why is it important for older adults to engage in resistance training. Existing position statements from leading professional organizations, such as the American College of Sports Medicine or the National Strength and Conditioning Association should be referenced here.
One comments on the study methodology. The authors state that the sample size was set at 8-10 peer mentors and 20-30 participants, but they provide no rationale for these sample size numbers. The authors provide results on 21 participants, which is on the low-end even for their own 20-30 participants range. Compared to most studies in the current literature on older adult resistance training, this is a low sample size. In fact, it's possible that the authors have arrived to non-significant statistical differences displayed in Table 2. due to the small sample size not providing them sufficient statistical power. The authors should acknowledge this methodological shortcoming in the Discussion section.
Author Response

(The authors gave the same response as above.)

Reviewer 3 Report
Dear Authors,
I think the article need to be improved, I have some suggestions:
Abstract.
To indicate the physical assessments done in the study
The conclusion does not fix with the title, even it have not been measured, the idea was to analyze if the participation in resistance training can improve with a peer, why the conclusion is that RT is safety? Safety was measured? How?
Main document
I suggest adding the abbreviation RT for “Resistance training”
Materials and method need to indicate de characteristics of the participants, numbers per group, etc… as well as the training characteristics, upper and lower body strength exercises, intensity, repetition, etc…
What means: “The sample size for this pilot study was set at 8-10 peers”? there were 21 participants
Why 14 at home and 7 al Gym? Why not 14 and 14?
To Include the meaning of all abbreviation in the tables
¿How home and gym are compared when the number of participants, session completed are different?
Abbreviations indicated in the document are not used, for example 6MWT or TUG (line 218…)
To be uniform. sometimes World Health Organization other WHO without including the meaning of the abbreviation in the document
Discussion must be improved, nothing is said about the results (6mWT, TUG….) the improvement is enough compare with other studies or not, to justified possible differents
Best Regards
Author Response

(The authors gave the same response as above.)
